# TGFβ1, MMPs and cytokines profiles in ocular surface: Possible tear biomarkers for pseudoexfoliation

**Prity Sahay**[1,2☯], **Shweta Reddy**[1], **Birendra Kumar Prusty**[3], **Rahul Modak**[2]\*, **Aparna Rao**[1,2☯]\*

**1** Hyderabad Eye Research Foundation (HERF), L.V. Prasad Eye Institute, Patia, Bhubaneswar, Odisha, India, **2** KIIT School of Biotechnology, Patia, Bhubaneswar, Odisha, India, **3** Institute of Life Science, Bhubaneswar, Odisha, India

☯ These authors contributed equally to this work.
\* rahul.modak@kiitbiotech.ac.in (RM); vinodini10375@yahoo.com (AR)

**Data Availability Statement:** All relevant data are within the paper and its Supporting information files.

**Funding:** This work was supported by the DBT/ Wellcome Trust India Alliance Fellowship IA/CPHI/

## Abstract

### Purpose

Pseudoexfoliation (PXF) is a unique form of glaucoma characterized by accumulation of exfoliative material in the eyes. Changes in tear profile in disease stages may give us insights into molecular mechanisms involved in causing glaucoma in the eye.

### Methods

All patients were categorized into three main categories; pseudoexfoliation (PXF), pseudoexfoliation glaucoma (PXG) and cataract, which served as control. Cytokines, transforming growth factor β1 (TGFβ1), matrix metalloproteases (MMPs) and fibronectin (FN1) were assessed with multiplex bead assay, enzyme-linked immunosorbent assay (ELISA), gelatin zymography, and immunohistochemistry (IHC) respectively in different ocular tissues such as tears, tenon's capsule, aqueous humor (AH) and serum samples of patients with PXF stages.

### Results

We found that TGFβ1, MMP-9 and FN1 protein expression were upregulated in tears, tenon's capsule and AH samples in PXG compared to PXF, though the MMP-9 protein activity was downregulated in PXG compared with control or PXF. We have also found that in PXG tears sample the fold change of TGF-α (Transforming Growth Factor-α), MDC (Macrophage Derived Chemokine), IL-8 (Interleukin-8), VEGF (Vascular Endothelial Growth Factor) were significantly downregulated and the levels of GM-CSF (Granulocyte Macrophage Colony Stimulating Factor), IP-10 (Interferon- γ produced protein-10) were significant upregulated. While in AH; IL-6 (Interleukin-6), IL-8, VEGF, IFN-a2 (Interferon- α2), GRO (Growth regulated alpha protein) levels were found lower and IL1a (Interleukin-1α) level was higher in PXG compared to PXF. And in serum; IFN-a2, Eotaxin, GM-CSF, Fractalkine, IL-10 (Interleukin-10), IL1Ra (Interleukin-1 receptor antagonist), IL-7 (Interleukin-7), IL-8, MIP1β

15/1/502031 awarded to Aparna Rao and the glaucoma foundation grant. The funders had no role in study design, data collection and analysis, decision to publish, or preparation of the manuscript.

**Competing interests:** The authors have declared that no competing interests exist.

(Macrophage Inflammatory Protein-1β), MCP-1 (Monocyte Chemoattractant Protein-1) levels were significantly upregulated and PDGF-AA (Platelet Derived Growth Factor-AA) level was downregulated in the patients with PXG compared to PXF.

## Conclusions

Altered expression of these molecules in tears may therefore be used as a signal for onset of glaucoma or for identifying eyes at risk of developing glaucoma in PXF.

## Introduction

Pseudoexfoliation (PXF) syndrome is a unique age related fibrillopathy characterized by mass deposition of white dandruff like material in different parts of anterior segment of the eye [1]. In our recent study, we elucidated different clinical correlates characterizing the different stages of pseudoexfoliation syndrome (PXF) and pseudoexfoliation glaucoma (PXG) with definite differences in tears of patients with PXG versus other forms of primary glaucoma [1, 2]. Vascular phenomenons such as vasospasm, systemic hypertension, angiographic vascular perfusion defects and oxidative stress markers such as generation of reactive oxygen species (ROS) have been found to be increased which partly explain the high incidence of ischemic ocular and systemic events associated with endothelial dysfunction seen characteristically in PXF [2–7]. Though PXF is well characterized clinically, the mechanism of onset of glaucoma in some eyes as well as the cause for such phenotypic diversity in PXF is unknown [1]. Several studies have highlighted the abnormal extra cellular matrix (ECM) homeostasis causing deposition of fibrillar material in the ocular tissues of PXF. This deposition may be triggered by various factors including genetic predisposition and environmental influences [4–7]. PXF is a protein misfolding disease which is characterized my dandruff like protein aggregates in different ocular structures. These aggregates are essentially composed of carbohydrate moieties with several ECM proteins like fibrillin and fibulin-5. Previous studies have demonstrated the composition of such aggregated proteinaceous accumulating material from enucleated eyes, which may not reflect the changes seen in earlier stages of the disease. The nature of aggregates of the aggregating material during disease evolution would be probably highlight molecular differences in each tissue depending on the local milieu. The TGF-β family is part of a superfamily of proteins which along with other cytokines and matrix metalloproteinases (MMPs) are key molecules implicated in altered ECM remodeling in PXF [1, 2, 8–11]. One study reported a negative correlation of TGFβ1 and TGFβ2 with lysine oxidase (LOX) activity in aqueous humor suggesting dysregulation of LOX activity despite elevated TGFβ levels in the aqueous [12]. While a significant positive correlation between TGFβ1 and presence of PXF and extent of angle pigmentation has been reported in another study, few other studies have showed no correlation of any clinical feature with onset of glaucoma [1, 13–17]. It is interesting to know how ECM remodeling is affected in each stage of the disease evolution and which molecules play a role in such a change. Paucity of procedures to measure ECM functions directly in patients representing earlier stages is the major obstacle. Levels of various cytokines have been found to correlate with the extent of IOP or glaucomatous disc damage in several independent studies [5, 8, 18]. Yet, the molecular mechanisms regulating clinical findings and those that cause transition from one phenotype to the other remain unclear.

TGFβ1, MMP-2, TIMP-2 levels in aqueous humor have been found to be elevated in PXG eyes as compared to POAG suggesting the role of TGβ1 in disease pathogenesis in PXG [2, 11,

12, 17]. MMP's are key molecules involved in TGFβ activation and a cascade of pathways leading to ECM degradation. Tear film's MMP levels have been found to correlate with disease progression in several eye diseases like diabetic retinopathy [19] though this remains largely unexplored in glaucoma.

While ECM functions can be discerned by aqueous or trabecular meshwork (TM) sampling in advanced stages of the disease with uncontrolled glaucoma requiring surgery, similar evaluation in earlier stages poses ethical concerns with an invasive sampling procedure. Tear analysis may offer a unique solution to understand disease pathogenesis in PXF wherein exfoliative deposits in tissue alterations have been seen in every ocular structure including the conjunctiva and tenons. This may mirror the changes in the aqueous, blood or trabecular meshwork which if compared can give useful clues to non-invasive evaluation of PXF eyes at risk of developing glaucoma. One study reported increased MMP-9 levels in the tear film in 80% PXF patients and 20% controls [10]. The AH or TM can be evaluated for molecular changes with severity of disease stage in pseudoexfoliation, sampling of the above is invasive requiring surgery which causes ethical concerns in early disease. Evaluating tear film TGFβ1, MMPs, FN1 and cytokines profiles would give us useful non-invasive methods of screening eyes at risk of developing pseudoexfoliation glaucoma. Our own study showed decreased tear MMP-9 activity in severe glaucoma and PXG eyes compared to other forms of primary glaucoma supporting an important role of MMP-9 in glaucoma onset in PXF. The aim of the current study was to evaluate MMPs, TGFβ1, FN1 and cytokines level in PXF and PXG patients to see the differences in their expression in between early and late stages of the disease. This project aimed to study the role of MMPs medicating ECM degradation and FN1 representing ECM production along with other proteomic signatures in clinical disease stages of PXF in different ocular tissues such as tears, tenons capsule, AH and serum samples.

## Materials and methods

### Patient screening and characterization

Study patients visiting the glaucoma service underwent a detailed evaluation including slitlamp evaluation, goldmann applanation tonometry, 4 mirror gonioscopy, +90D fundus biomicroscopy, tear break uptime, Schirmers test (without anesthesia), humphrey visual fields. This study was approved and conducted in compliance by the institutional ethics committee of LV Prasad Eye Institute, MTC Campus, Bhubaneswar, India and adhered to the tenets of the Declaration of Helsinki. A written informed consent was obtained from all patients after explanation of the nature and possible consequences of the study. Inclusion criteria for pseudoexfoliation syndrome, its clinical variants and stages are detailed elsewhere [1, 2]. Briefly pseudoexfoliation included newly diagnosed patients and the diagnosis was done in this study. Patients with evident classical dandruff or flaky deposits on the pupil, lens or other ocular structures, radial pigment over the lens surface with or without raised intraocular pressure were present in this study. Only newly diagnosed medically naïve patients were recruited for the study to avoid bias in cytokines profiles due to chronic use of anti-glaucoma drugs and drug induced dry eye. Glaucoma was defined as those with glaucomatous optic neuropathy evident by disc changes like cupping, rim thinning or defects, focal notch or retinal nerve fiber layer defects with corresponding visual field defects. Both unilateral and bilateral cases were included in this study where only affected eyes (defined above in inclusion criteria) were considered as cases. Patients with uveitis, neovascular glaucoma, past laser procedures or anti-glaucoma medical treatment, conjunctivitis, allergic blepharitis or dry eye and clinically uninvolved eye were excluded. Patients with any other autoimmune or neurodegenerative disorder and diabetes mellitus developing at any time during the study were also excluded. Visual

field defects were classified as glaucomatous if glaucoma hemifield test outside normal limits or pattern standard deviation with probability <5%, which were reproducible over three baseline fields.

The eyes were further classified into stages with PXF only comprising the earliest stage with clinically evident form of PXF (as described above) without raised IOP and normal optic nerve/visual field. PXG included those with disc and field changes consistent with diagnosis of glaucoma.

Non-glaucomatous control subjects comprised of patients recruited from general eye clinic with no history of any ophthalmic disease or abnormality with no history of topical medical treatment previously. Patients with a known diagnosis of dry eyes, diabetes mellitus, systemic autoimmune disease or those with evidence of intraocular inflammation from any cause were excluded from the study.

After characterizing patients into different clinical stages as above, samples were collected from patients included for the study.

## Sample collection

Tears were collected from outer canthus of patients by the same technique described previously elsewhere [2]. Tear flo™Schirmer filter strip was used to collect tears from PXF cases which included pseudoxfoliation (PXF), pseudoexfoliation with glaucoma (PXG) and cataract as control. After collecting written consent, PXF patients were asked to sit in upright position looking straight after which tear strips were placed in the inferior fornix for 5 minutes under aseptic conditions without topical anesthetic to collect tears. Aseptically the tear strips were collected in 2ml microcentrifuge tube and the procedures of the isolation of protein from the schirmer strip were followed as previously described [2]. Total protein in each sample was assessed using Bradford assay (Biorad, California, United States).

Tenons capsule harvested from the superior quadrant surgery (before any incision) from patients undergoing either glaucoma or cataract surgery for immunohistochemistry or immunofluorescence experiments.

Aqueous humor (AH) collection was done intraoperatively using a 1 cc syringe before entering the eye for any other procedure and immediately aliquoted and stored at -80C for further analysis.

Bloods were collected from PXF and PXG patients and undergoing cataract surgery using standard aseptic precautions. To isolate serum, blood from different study patients were collected in vials without any coagulants and stored at room temperature for half an hour. It was then centrifuged at 8000 rpm for 15 minutes at 4˚C, the sera was collected, aliquoted in different vials to prevent repeatedly freeze thaw and stored at -80˚C for later ELISA assays.

## Enzyme linked immunosorbent assay

Concentration of transforming growth factor (TGFβ1) was analyzed in different PXF/PXG/Control samples of tears (n = 6) and aqueous (n = 4) by TGFβ1 ELISA Kit as per the guidelines described in R&D System. Concentrations of Matrix metalloproteases (MMP-1) and tissue inhibitors of metalloproteases (TIMP-1) from 10 pooled samples of tears in one (n = 20) tears samples from each group and individual serum samples (n = 20) were analyzed by colorimetric immunoassays performed according to the instructions of the manufacturers (ThermoScientific, Massachusetts, United States). All procedures were preformed according to the ELISA instructions and the manufacturers' instructions. Finally, absorbance was measured on an ELISA microplate spectrophotometer (Epoch, Biotek, USA) set at 450nm and 550nm. To

minimize optical imperfections in the microplate, values from 550nm were subtracted from 450nm.

## Gelatin zymography for MMPs profiles

Gelatin zymography was used to access the functional activities of MMP-9 and MMP-2 expression in tears, serum and aqueous humor patients with different stages of pseudoexfoliation syndrome. Tear samples of PXF (n = 50), PXG (n = 50) and cataract (n = 50) as control were taken for the analysis. Serum and AH samples were taken from PXF samples (n = 10), PXG (n = 10) and controls (n = 10). Equal amount of proteins obtained from these samples and controls were separated on 10% SDS-PAGE gels containing 0.1% gelatin. Then the protocols were strictly followed as previously described [2]. The zymographic gels were imaged and lysis/digested zones in every lane was analyzed using image J software (http://imagej.nih.gov/ij/; provided in the public domain by the National Institutes of Health, Bethesda, MD, USA) to obtain band intensity with MMP-2 and -9 activities expressed in arbitrary units (A.U).

## Immunohistochemistry

Samples of human tenon's capsule from PXF (n = 5), PXG (n = 5) and control (n = 5) harvested from the superior quadrant surgery (before any incision) from patients undergoing either glaucoma or cataract surgery. These samples were analysed for TGFβ1, MMP-9 and SMAD-3 expressions. Protocol of immunohistochemistry was followed strictly as previously described [2]. The sections ware incubated with mouse primary anti-MMP-9 antibody (1:100, MAB13458; Merck Millipore), anti-TGFβ1 (1:150, ab27969; abcam) and anti-SMAD-3 antibody (1:500, ab40854; abcam) at 4° C for overnight. After washing with PBS, sections were incubated with biotinylated secondary goat anti-mouse antibody (Dako) for 30 minutes. Immune reactions were visualized with incubation in 3, 3'Diaminobenzidine tetrahydrochloride (DAB) for 8 minutes in dark following the manufacturer's protocol (LSAB2 System-HRP, Dako). Sections were counterstained with hematoxylin and fixed subsequently. Slides were examined under a bright field (CKX53, Olympus, Tokyo, Japan) microscope at ×20 and ×40 magnification and images were analyzed by MagVision software.

In each case, the percentage of TGFβ1, MMP-9 and SMAD-3 positive staining of each slide in triplicates was determined. Tissue sections were scored by determining the proportion of stained extracellular matrix near cells relative to the overall extracellular matrixes. Evaluation of TGFβ1, MMP-9 and FN1 expressions were performed at x40 magnification. The distribution (%) of each antibody was evaluated according to the following criteria: 0 (<5), 1 (6–25), 2 (25–50), 3 (51–75), and 4 (>75) of cells displaying positive immunoreactivity. The intensity of antibody immunostaining was scored as: 0 (none), 1 (weak), 2 (moderate), and 3 (strong). Total immunostaining scores, which combined intensity and distribution of immunostaining, were ranked as low (intensity 0–1 and distribution 0–4, or intensity 2 and distribution 0–1) or high (intensity 2–3 and distribution 2–4) [20, 21].

## Cytokine analysis

Different cytokines in the extracted protein samples from 10 pooled samples of tears in one (n = 20), individual samples of serum and aqueous humor from each group (n = 3) were evaluated using a convenient bioplex kit assay (Milliplex MAP kit, HCYTMAG-60K-PX41, Millipore, Massachusetts, United States). Detail of cytokines was given in S1 Appendix. Briefly, proteins from tears samples from different groups of PXF were extracted as described earlier in the method section of protein extraction from the tears. Proteins from same group were

pooled and assessed further for the cytokine analysis as described in the manufactures protocol.

The plates were run on the Bio-Plex® 200 system (Luminex Corporation, Texas, USA). Before each assay run, the system was calibrated with the Bio-Plex® calibration kit and validated with the Bio-Plex® validation kit 4.0. Bio-Plex® sheath fluid served as the delivery medium for the samples. Analysis was performed with Bio-Plex® manager 6.1 the software used was *xPONENT* software. Within the device settings, 50 events per bead region were defined as minimum criterion.

## Western blot analysis

To identify the differential expressions for fibronectin (FN1) in tears samples of different cases and control of pseudoexfoliation samples (n = 3), western blot analysis was performed using their specific antibodies. A total of 40ng of tears proteins from tears lysates were separated by 12% SDS-PAGE electrophoresis under denaturing and reducing conditions. Protein bands were transferred to a polyvinylidenedifluoride (PVDF) membrane, which was then blocked with 5% fat-free milk in tris buffered Saline (TBS), pH 7.5, followed by incubation with anti-FN (1:1000, ab6328; abcam) overnight at 4˚ C with continuous shaking. Blots were washed with 1% TBST and then incubated with the respective secondary antibody (1:3000, abcam) for 2 hours at room temperature. GAPDH was used as the protein loading control. Immunoreactive proteins were detected with Clarity Western luminol/enhancer and peroxidase solution (Bio-rad, United States). The protein levels were determined by densitometry with Image J (National Institutes of Health, Bethesda, MD).

## Immunofluorescence

The Tenon's capsule from pseudoexfoliation patients (n = 3) of each cases and controls were sectioned to 6µM thickness embedded in paraffin and deparaffinized in xylene and rehydrated in a series of graded ethanol solutions. Tissues were fixed with 4% paraformaldehyde for 20 minutes. The sections were washed thrice in 1X PBS for 5 minutes. Next, these sections were permeabilized with 1% Triton X-100 in PBS for 15 minutes and blocked with 1% BSA in PBS for an hour at 4˚ C before being incubated in the primary antibody against FN1 at 1:150 (ab6328, abcam) overnight at 4˚ C. After three washes with PBS, the samples were incubated with corresponding AlexaFluor–conjugated secondary IgG at 1:500 for 60 minutes. The samples were then mount with slow FADE gold and anti-FADE DAPI and analyzed with a Zeiss ApoTome.2 florescence microscope (Carl Zeiss, Germany).

## Quantification and statistical analysis

Image J was used to assess protein band intensities on gelatin zymographic gels and western blot. For all average intensities per pixel values were recorded, rectangular areas along the bands were selected with a width of 5 pixels. Background intensities were also subtracted. Data from each experiment were analyzed using Graph-Pad Prism with column and grouped comparison. Results were presented as means ± standard error of mean (SEM) of triplicate experiments. Data were analyzed by ANOVA post hoc t test with Tukey correction with P value of <0.05 considered significant.

## Results

In this study total number of 360 eyes from different patients out of which 180 cases and 78 controls were examined which included PXF (n = 120), PXG (n = 120), and control (n = 120).

**Table 1. Demographic and clinical characteristics of patients with psuedoexfoliation included in the study.**

| | Controls (N = 120) | PXPXF PXF (N = 120) | PXG (N = 120) | P value |
|---|---|---|---|---|
| Age (years) | 58±6.8 | 62±5.8 | 64±8.1 | 0.7 |
| Male:Female | 62:38 | 58:42 | 73:27 | 0.6 |
| Mean Deviation (dB) | 1±1.1 | 2±3.2 | -10±6.8 | <0.001 |
| Baseline IOP (mm Hg) | 12±3.2 | 14±3.1 | 26±3.4 | <0.001 |

IOP-Intraocular pressure, PXF-pseudoexfoliation, PXG-pseudoexfoliation glaucoma.

Table 1 showed significant difference in mean deviation between PXF and PXG (P<0.001) with no statistical difference in age of the patients between stages (P = 0.7). It was noticed that the mean baseline IOP differed significantly between cases and controls, with the maximum increase of IOP found in PXG cases than control (Table 1, P<0.001).

## Expression/activity of TGFβ1, MMPs/TIMPs in tears, tenon's capsule in different stages of pseudoexfoliation

TGFβ1 protein levels were found to be high in PXG when compared with PXF cases or control by ELISA (Fig 1a). Gelatin zymography revealed increased MMP-9 activities in earlier forms of the disease with reduced activity in PXG eyes which are comparable to controls (Fig 1b and 1c). No significant changes were observed in the activity of MMP-2 in any stages, (Fig 1b). We next evaluated the level of MMP-1, TIMP-1 expression using ELISA in pooled tear samples. Fig 1d, shows the expression of MMP-1 was significantly upregulated in PXF compared to control then lowers in PXG while the differences in the expression of TIMP-1 was not changed in the stages of PXF.

We further wanted to see expressions of TGFβ1 and MMP-9 in tenons capsule in different stages of pseudoexfoliation samples by immunohistochemistry. Strong staining of TGFβ1 was observed for PXG, the TGFβ1 expression was maximal in the endothelial space in early stages, and eyes with glaucoma had maximal expression in epithelial or sub-epithelial layers (Fig 1e, Table 2). On the other hand, the protein expression of MMP-9 was higher in PXG when compared with PXF or control. We also found that maximum of MMP-9 is localized in extracellular spaces of sub epithelial layer in tenons capsule, (Fig 1e, Table 2).

## Expression/activity of TGFβ1, MMPs/TIMPs in aqueous humor and serum in different stages of pseudoexfoliation

We also evaluated the concentration of TGFβ1 by ELISA in aqueous samples; and found that the concentration of TGFβ1 was more in PXG which is parallel with tear results, (Fig 2a). MMP-9 activities also paralleled the findings in tears with increased expression in PXF compared to PXG or controls (Fig 2b and 2c).

Concentration of TGFβ1 was evaluated in our previous study [20] and as expected these results are consistent with our previous results as we found in tears and AH, its expression is higher in PXG as compared to PXF or control. The MMP-9 and MMP-2 activities in serum were significantly upregulated in PXF and PXG compared to control while MMP-9 significantly downregulated in PXG when compared to PXF (Fig 2d and 2e). TIMP-1 and MMP-1 expression were initially upregulated in PXF and then downregulated in PXG compared to control (Fig 2f).

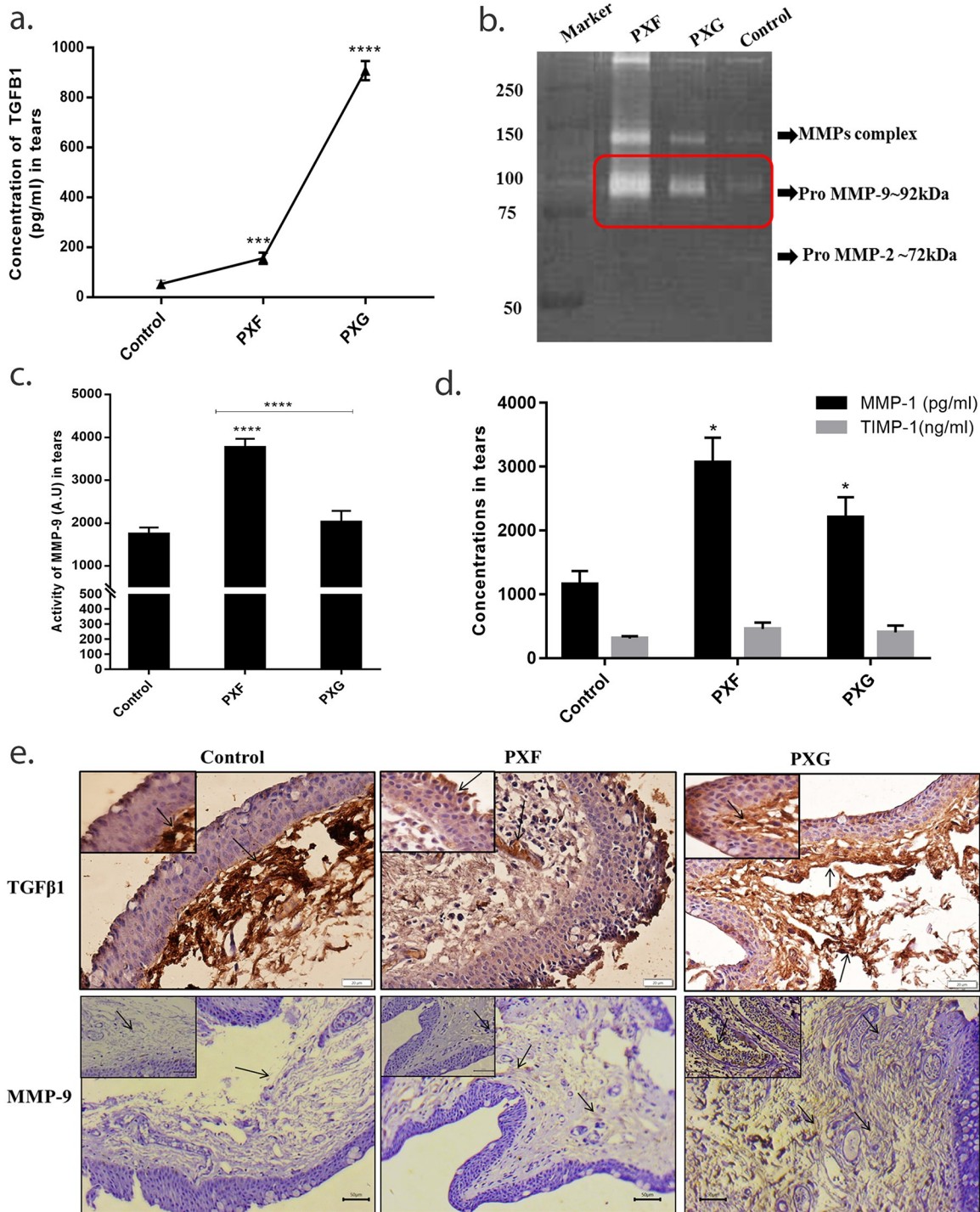

**Fig 1. Expression/activity of TGFβ1, MMPs/TIMPs in tears, tenon's capsule in different stages of pseudoexfoliation.** a, concentration of TGFβ1 in tears (n = 6) using enzyme linked immunoassay; b and c, representative image and graph showing expression of MMP-9 in tears (n = 50) using gelatin zymography; d, concentration of TIMP-1 and MMP-1 in the 10 pooled tears in one (n = 20) by enzyme linked immunoassay; e, bright field microscopic images at ×20 and ×40 (inset) magnifications of TGFβ1 and MMP-9 in tenons capsule (n = 5) using imunohistochemical study. Scale bar-20um, 50um, Means ± SEM shown, *p<0.05, ***p<0.001, ****p<0.0001 one way ANOVA post hoc t test with Turkey correction, PXF- pseudoexfoliation, PXG- pseudoexfoliation glaucoma.

**Table 2. The immunostaining scores of TGFβ1, MMP-9 and SMAD-3 in human tenonscapsule.**

| Controls/ Cases | Distribution of (%) | | | Intensity | | | Total immunostaining scores | | |
|---|---|---|---|---|---|---|---|---|---|
| | TGF-β1 | MMP-9 | SMAD-3 | TGF-β1 | MMP-9 | SMAD-3 | TGF-β1 | MMP-9 | SMAD-3 |
| Controls | 2 (26–50%) | 1 (6–25%) | 3 (51–75%) | 3 | 0 | 3 | high | low | high |
| Pseudoexfoliation | 3 (51–75%) | 2 (26–50%) | 1 (6–25%) | 3 | 1 | 0 | high | low | low |
| Pseudoexfolation glaucoma | 4 (>75%) | 4 (>75%) | 1 (6–25%) | 3 | 3 | 0 | high | high | low |

## Cytokines/chemokines profile in tears, aqueous humor (AH) and serum samples

Changes in cytokine levels may be suggestive of progression of disease. In tears, notably, the fold change of TGF α, MDC, IL-8 and VEGF were significantly downregulated in PXG while GM-CSF and IP-10 were significantly upregulated in PXF when compared to control (Fig 3a).

While the level of cytokines in AH report that the fold change of IFNa2, GRO, IL-6, Il-8 and VEGF levels were significantly downregulated in PXG while the levels of IL1a was found maximally in PXG eyes (Fig 3b).

The levels of fold change of EGF-2, Eotaxin GM-CSF, Fractalkine, IFN-a2, IL-10, IL-Ra, IL-7, IL-8, MCP-1, and MIP1β were noticed to be significantly lower in PXG when compared to the levels of PXF samples (Fig 3c). We have also seen that PDGF-AA was upregulated in PXG when compared to PXF.

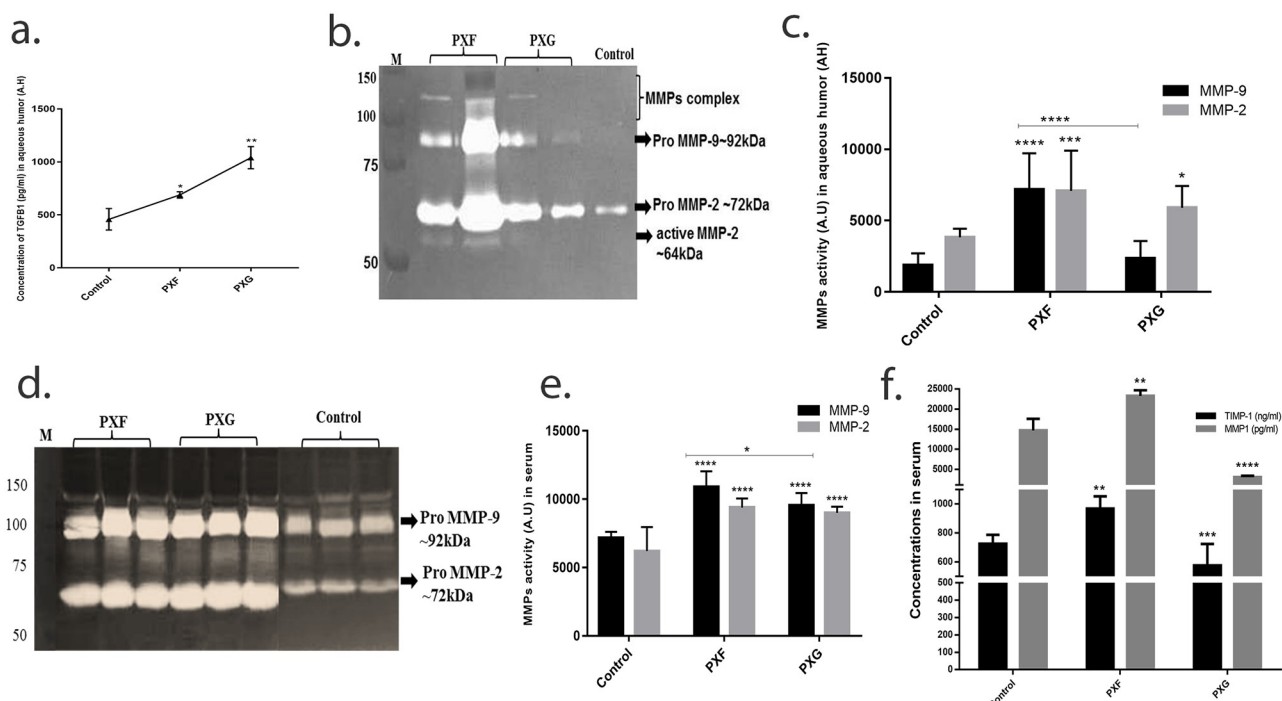

**Fig 2. Expression/activity of TGFβ1, MMPs/TIMPs in aqueous humor and serum in different stages of pseudoexfoliation.** a, concentration of TGFβ1 in aqueous humor (n = 4) using enzyme linked immunoassay; b and c, representative image and graph showing expression of MMP-9 in aqueous humor (n = 10) using gelatin zymography; d and e, representative image and graph showing expression of MMP-9 in serum (n = 10); f, concentration of TIMP-1 and MMP-1 in the and serum samples (n = 20) by enzyme linked immunoassay; Means ± SEM shown, *p<0.05, **p<0.01, ***p<0.001, ****p<0.0001, one way ANOVA post hoc t test with Turkey correction, PXF- pseudoexfoliation, PXG- pseudoexfoliation glaucoma.

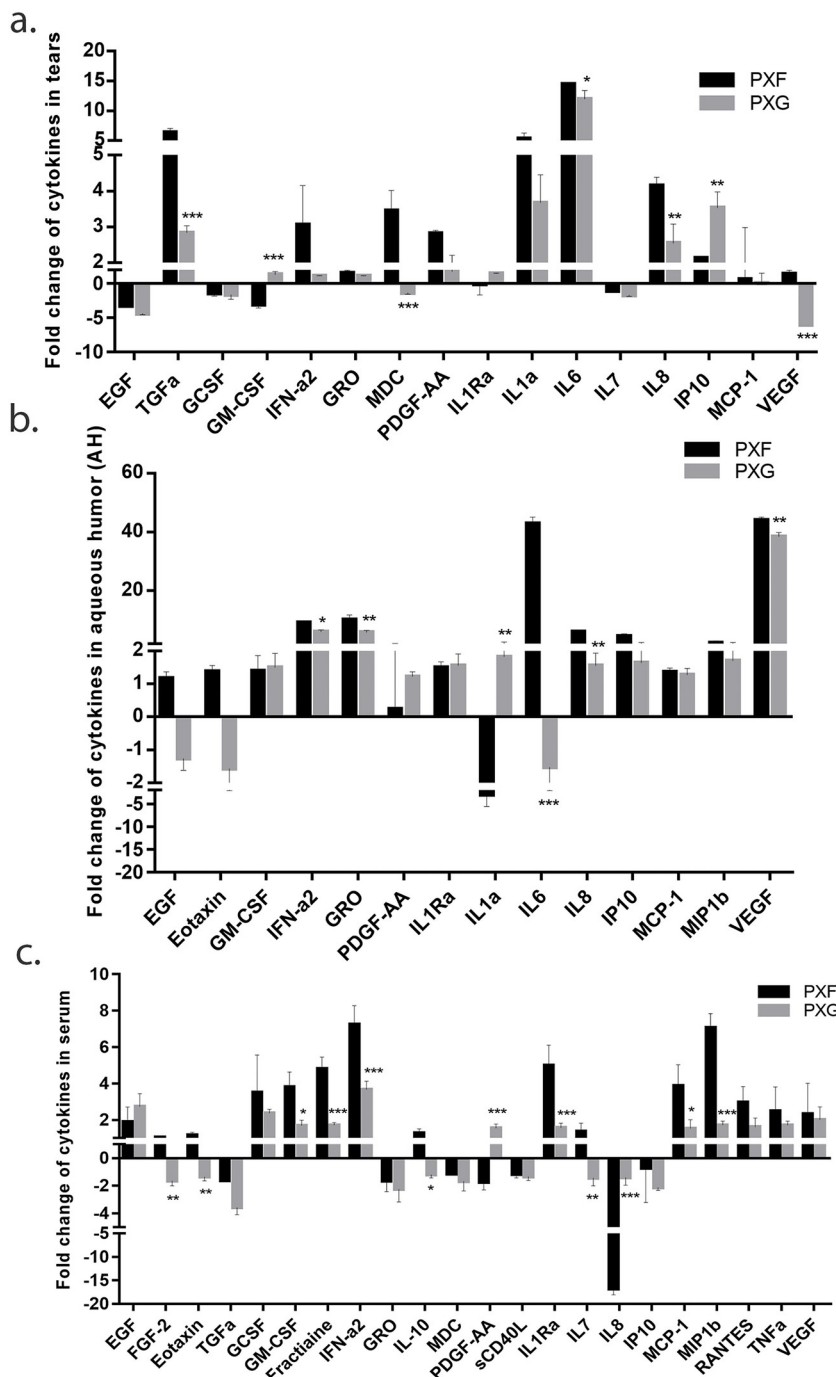

**Fig 3. Cytokines/chemokines profile in tears, aqueous humor and serum samples.** Graph shows the fold change of cytokines in 10 pooled tears in one (n = 20) aqueous humor (n = 3) and serum samples (n = 3) by multiplex bead assays. PXF- pseudoexfoliation, PXG- pseudoexfoliation glaucoma. Means ± SEM shown, *p<0.05, **p<0.01, ***p<0.001, one way ANOVA post hoc t test with Turkey correction, PXF- pseudoexfoliation, PXG- pseudoexfoliation glaucoma.

### Evaluation of FN1 in ocular tissue

Previous studies have shown that FN1 acts as a major regulator of ECM synthesis and deposition [21, 22]. Our results showed that there is an increase of TGFβ1 and MMP-9 which was consistent in each ocular tissue in the later stages of PXF which may have role in the pathogenesis of glaucoma. So we wanted to know that whether cellular FN1, an insoluble fibronectin isoform that forms fibril, found supporting results which may help in regulating ECM-cell interaction.

For this, FN1 expression was evaluated by immunoblotting in tears of PXF, PXG and control cases, we found a strong band near 250-260kDa (S1 Raw images), which confirmed that the protein expression of soluble FN1 was lower in PXG than compared to PXF or control (Fig 4a and 4b). To confirm its expression, immunostaining of tenon's capsules were performed, we found over expression of FN1 in all stages of PXF compared to controls, its expressions were found to be localized to the ECM space suggesting an increase in insoluble fractions as disease severity progresses, (Fig 4c). This was also associated with decreased expression of SMAD-3 despite defined increased TGFβ expression in the tenons suggesting FN1 or MMP-9 regulation by non-canonical TGFβ signaling pathway in the ECM proteins production (Fig 4d, Table 2).

## Discussion

In this study we found TGFβ1 and MMP-9 protein concentrations were high in PXG but MMP-9 activity was reduced in eyes with glaucoma. The cytokine signatures were different in different clinical disease stages with PXF being characterized by maximal elevation of inflammatory cytokines. The FN1 expression seems to suggest an increased ECM production in severe stages which represented an insoluble form in protein aggregates which was however not found to be correlated with increased TGF expression suggesting an alternate regulation of ECM production in severe disease stages.

PXF is known to be an ECM disorder with inconsistencies of genetic or proteomics signatures in different ethnic populations [2, 3, 7, 8, 14, 23–30]. Fibronectin (FN1) is one of the predominant ECM proteins and it is known to regulate the assembly of other ECM proteins in trabecular meshwork (TM) cells. It has been shown that the ligation of the FN1 receptor induced the gene expression of both collagenase and stromelysin by fibroblasts [31, 32]. Few studies suggest that the increase of IOP is believed to be caused by excessive deposition of FN1 in ECM of TM tissue. For example, Baneyx et al demonstrated the conformation of highly extended fibrils of FNI in matrices, confirmed by fluorescence resonance energy transfer (FRET) [33]. Chen Y and their group reported that the complexity of FN1 would affect IOP by changing the basic physiology of cells properties or by the change of the events of cell-matrix signaling. Furthermore their group demonstrated FN1 by the use of high resolution Cryo-SEM studies, this support that *in-vitro* they found that some fibrils were very straight whereas others were highly nodular and coiled [34]. Since FN1 regulates the functioning of these above systems by modulating cell-matrix protein binding, aberrant regulation of integrin function and ECM homeostasis induced by FN1 in the TM in PXG may be worth exploring.

MMPs play an important role in ECM homeostasis in the eye and imbalance in MMP regulation and function is postulated as one of the key molecular disturbances characteristic of PXF [2, 3, 7, 9, 10, 14]. The ECM and the TM cell form a matrisome complex with intricate balance and cross-talk between several molecules regulating ECM production and ECM degradation required for TM cell function. Close knit relation between the TM cell and the ECM holds the key to abnormal ECM degradation which leads to deposits of exfoliation material around the TM and in different ocular structures which further impair tissue function causing

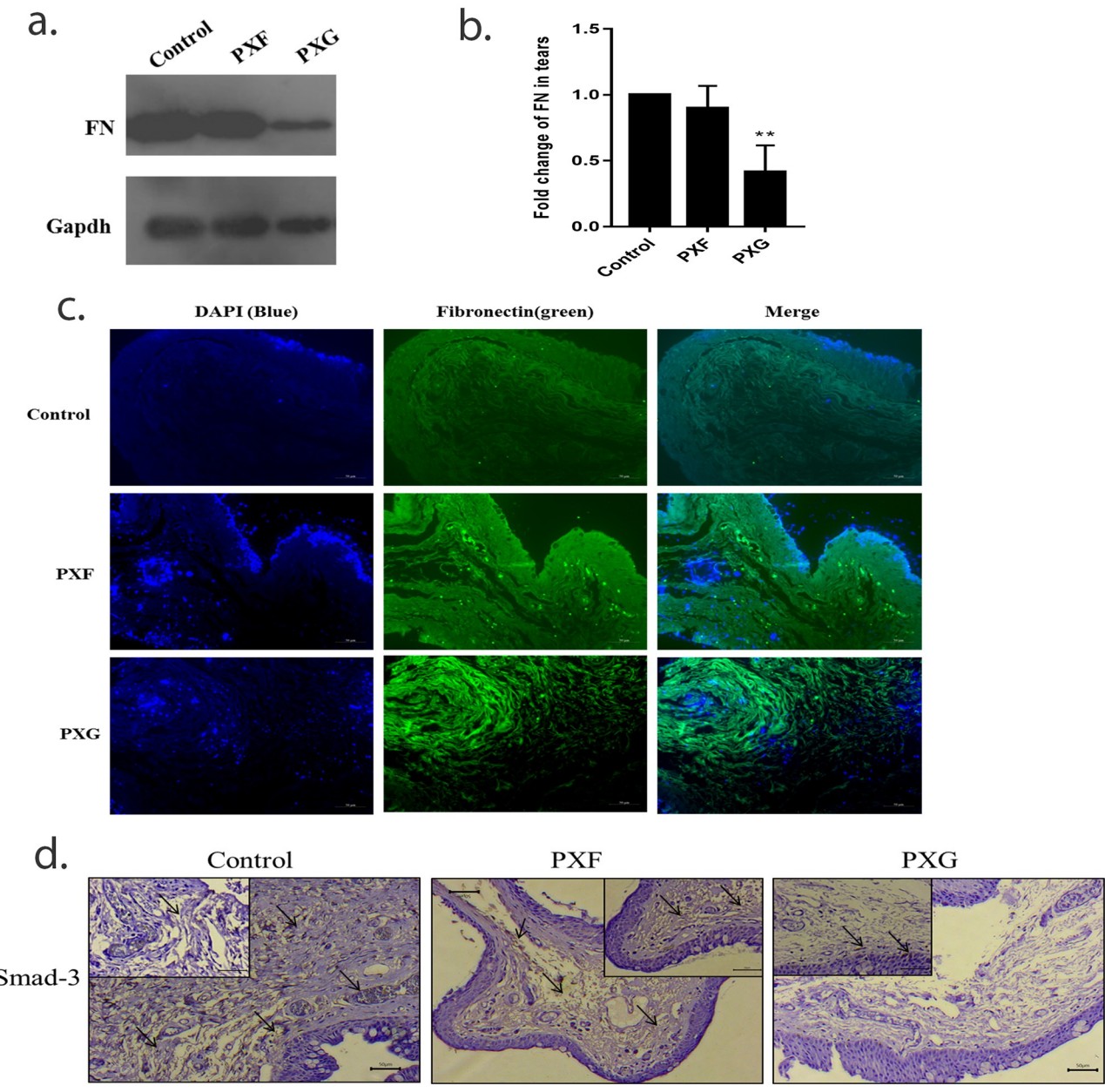

**Fig 4. Evaluation of FN1 in ocular tissue.** a and b, protein level of FN1 in tears (n = 3) using western; c, immunofluorescent images of FN1 in tenons capsule (n = 5) at ×20 magnification; d, bright field microscopic images at ×20 and ×40 (inset) magnifications of SMAD-3 in tenons capsule (n = 3) using imunohistochemical study. Scale bar-10um, Means ± STDEV shown, **p<0.01, one way ANOVA post hoc t test with Turkey correction, PXF-pseudoexfoliation, PXG- pseudoexfoliation glaucoma.

glaucoma. Earlier studies have reported higher levels of TIMP in aqueous or TM in patients with PXG and other forms of glaucoma [10]. The levels of MMP-2 also have been reported to be higher among other MMPs with imbalance between MMP and TIMP levels in the aqueous being responsible for accumulation of exfoliation deposits owing to reduced or aberrant ECM degradation while our earlier study did not find elevated MMP-2 in PXF eyes [2, 10]. MMP-9 is one such key molecule involved in ECM degradation and its role as a tear-based predictive marker for glaucoma has already been suggested in our previous study [2]. We found the

activity of MMP-9 getting significantly reduced in PXG compared to other forms of glaucoma suggesting exhaustion of degradatory mechanisms in late stages of the disease. Elevated MMP-9 activity in earlier disease stages and reduced activity despite TGFβ elevations suggests alternate regulation of MMP-9 activity in advanced stage of PXG despite high MMP-9 protein levels in the local mileu which may be the key for reduced or aberrant ECM degradation and TGFβ downstream pathway dysregulation seen in later disease stages. In contrast, we did not find any difference in the MMP-2 or TIMP-1 levels in different stages or variants of the PXF. We believe that these differences in MMP-9 and MMP-2 levels in this and earlier studies may reflect differences owing to stringent disease staging and phenotyping into clinical variants which has largely been overlooked. How activated MMP-9 is regulated in each stage of PXF may hold the key for correcting the aberrant ECM deposition and fibrosis seen in later stages causing glaucoma. Elevated FN1 levels with reducing MMP-9 activity despite increased TGF levels in severe disease stages suggest an escape of the downstream ECM related pathways and alternate control by non-canonical pathways as glaucoma sets in.

We observed higher levels of cytokines in pigmentary form of the disease, which is characterized by radial pigments on the lens signaling early exfoliation deposits. This form has been referred to as the pre-capsular form of PXF and is traditionally thought to represent an earlier form of the disease [1, 4, 27]. Yet, in our earlier study, we did find eyes with this form also requiring medicines for IOP control suggesting that these eyes also may develop OHT or glaucoma and contrary to conventional belief does not represent earlier stage of the disease [1]. Breakdown of blood-aqueous barrier is a well-known phenomenon in PXF and is believed to occur in PXF even before clinically evident deposits or glaucoma [3, 7, 27]. Such changes in the iris vasculature coupled with frequent rubbing of the iris with the lens may be responsible for the radial pigments or pattern of deposits seen in PXF [1]. These observations suggest that the iris may be the source of PXF material in the anterior segment. The observation of elevated cytokines in pigmentary form of the disease as seen in this study may indicate that this form is associated with earlier breakdown of blood aqueous barrier owing to release of iris pigment causing a higher inflammatory milieu which may be the predominant mechanisms of disease pathogenesis in this form of PXF compared to classical PXF. The ubiquitin proteasomal system is known to maintain vascular homeostasis by affecting endothelial nitric oxide synthase activity [35]. Altered endothelial functioning as observed in the TM region of glaucoma patients can be directly correlated with the series of events regulated by the ubiquitin proteosomal system [36]. The destructive role of TNF pathway in various retinal diseases and disorders like glaucomatous neurodegeneration is well documented [37–39]. Also, the pathogenesis of the disease in relation with the immune system is mainly due to the complex interplay between hypoxia, oxidative stress, autoimmune processes and other factors [38, 40].

This study did not include unaffected eyes without the evidence of PXF deposits which may signal the earliest form of the disease. We also excluded eyes on medical treatment which may confound cytokine or protein signatures; therefore application for the results of this study cannot be used for eyes on medical therapy. We are unsure if these signature or cytokine profiles can be generalized to all ethnic populations. Nevertheless, we believe that novel proteins and cytokine signatures indicate a common mechanistic difference in different stages of the disease.

In summary, altered expression of downstream molecules regulating extra cellular matrix (ECM) degradation (MMP-9) and production (FN1) controlled by TGFβ suggest dysregulation, however, an escape of the TGF control on ECM homeostatic mechanisms as PXF disease progresses. Altered expression of these molecules may therefore be used as a signal for onset of glaucoma or for identifying eyes at risk of developing glaucoma in PXF.

## Supporting information

**S1 Raw images. Uncropped and unadjusted zymographic gel images and full blots of FN and Gapdh.** Zymographic gel image of (a) tears; (b) aqueous humor; (c) serum; (d & e) blot of FN and Gapdh; lane 1, control; lane 2, PXF; lane 3, PXG.
(PDF)

**S1 Appendix. Full forms of abbreviated cytokines used in bioplex kit assay.**
(DOC)

## Author Contributions

**Conceptualization:** Aparna Rao.

**Data curation:** Prity Sahay, Birendra Kumar Prusty, Aparna Rao.

**Formal analysis:** Prity Sahay, Birendra Kumar Prusty, Aparna Rao.

**Funding acquisition:** Aparna Rao.

**Investigation:** Prity Sahay, Aparna Rao.

**Methodology:** Prity Sahay, Shweta Reddy.

**Project administration:** Aparna Rao.

**Resources:** Aparna Rao.

**Software:** Birendra Kumar Prusty, Aparna Rao.

**Supervision:** Rahul Modak, Aparna Rao.

**Validation:** Aparna Rao.

**Visualization:** Aparna Rao.

**Writing – review & editing:** Prity Sahay, Shweta Reddy, Rahul Modak, Aparna Rao.

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
