## [Decision Letter · Decision Letter 0]

3 Mar 2021

PONE-D-21-01128

TGFβ1, MMPs and cytokines profiles in ocular surface: Possible tear biomarkers for pseudoexfoliation

PLOS ONE

Dear Dr. Rao,

Thank you for submitting your manuscript to PLOS ONE. After careful consideration, we feel that it has merit but does not fully meet PLOS ONE’s publication criteria as it currently stands. Therefore, we invite you to submit a revised version of the manuscript that addresses the points raised during the review process.

We look forward to receiving your revised manuscript.

Kind regards,

Andrew W Taylor, Ph.D.

Academic Editor

PLOS ONE

Journal Requirements:

https://www.mdpi.com/2073-4409/8/12/1518/html (discussion section)

https://iovs.arvojournals.org/article.aspx?articleid=2630993 (beginning of results section)

In your revision ensure you cite all your sources (including your own works), and quote or rephrase any duplicated text outside the methods section. Further consideration is dependent on these concerns being addressed.

Reviewers' comments:

Reviewer's Responses to Questions

**Comments to the Author**

1. Is the manuscript technically sound, and do the data support the conclusions?

Reviewer #1: Yes

2. Has the statistical analysis been performed appropriately and rigorously? 

Reviewer #1: Yes

3. Have the authors made all data underlying the findings in their manuscript fully available?

Reviewer #1: Yes

4. Is the manuscript presented in an intelligible fashion and written in standard English?

Reviewer #1: No

5. Review Comments to the Author

Reviewer #1: The manuscript reports changes in tear profile in Pseudoexfoliation for onset of glaucoma. It is suggested that authors revise English and avoid using complex sentences in addition to clarifying some of the points listed below:

Lines 51-52 - Instead of asking questions please define these topics for the reader at the outset.

Line 56 - increased in what?

Line 57 - which disease? Try to write in simple clear sentences.

Lines 57-58 - 'Though this disease' which disease?

Line 59 - which entity?

Line 62 - what does 'This' refers to?

Line 62-67 - Complex sentence. Please write simple sentences.

Line 69 - 'would be probably hold differences' - not clear

Line 78 - what or which?

Lines 103-104 - Write in simple sentences - The aim of the current study was...

Line 105 - 'In brief' - this is rather explanatory than brief

Line 120 - please specify if the diagnosis was done in this study

Line 153 - procedure for what was followed?

Line 170 - which guideline? Writing name of the manufacture in brackets will be good.

Line 201 - 'examined examination' - correct the sentence

Table 1 - Any reasoning for high male vs female numbers?

Line 312 - 'surprisingly' - state what was the expectation otherwise?

Line 402 - 'Earlier studies' - mention the reference

Line 436 - 'release of iris pigment release' - correct the sentence

6. PLOS authors have the option to publish the peer review history of their article (what does this mean?). If published, this will include your full peer review and any attached files.

Reviewer #1: No

---

## [Author Response · Author response to Decision Letter 0]

16 Mar 2021

Author's Response To Reviewer Comments:

Journal Requirements:

Author response: We have checked the journal requirement and my manuscript meets PLOS ONE’s style requirements.

2. PLOS ONE now requires that authors provide the original uncropped and unadjusted images underlying all blot or gel results reported in a submission’s figures or Supporting Information files.

Author response: We have already added original uncropped and unadjusted blot images in supplemental information as S1 Figure in my original manuscript and in the revised manuscript we have added uncropped and unadjusted gel image as S2 Figure (legend was added in the end of manuscript where Supplemental information is given) present in the supplemental information. Further we ensure that my figures in the result section adhere fully to journal guidelines and we provided the original underlying images for all blot or gel data reported in my submission.

Author response: In our revise manuscript I cited all other sources (including your own works), and rephrased the duplicated texts in the beginning of result (line no 262-267) and discussion (line no 385-391) section which is highlighted. 

Reviewers' comments:

Review Comments to the Author

Reviewer #1: The manuscript reports changes in tear profile in Pseudoexfoliation for onset of glaucoma. It is suggested that authors revise English and avoid using complex sentences in addition to clarifying some of the points listed below:

Author response: We have professionally edited the manuscript for grammar and used simple sentences as suggested.

Lines 51-52 - Instead of asking questions please define these topics for the reader at the outset.

Author response: The sentences have been rephrased and defined the disease for better understanding.

Line 56 - increased in what?

Author response: “Vascular phenomenon such as vasospasm, systemic hypertension, angiographic vascular perfusion defects and oxidative stress markers such as generation of reactive oxygen species (ROS) have been found to be increased which partly explain the high incidence of ischemic ocular and systemic events associated with endothelial dysfunction seen characteristically in PXF”. The sentence is changed as per your suggestion in the revised manuscript.

Line 57 - which disease? Try to write in simple clear sentences.

Author response: The sentence is changed as per your suggestion in the revised manuscript.

Lines 57-58 - 'Though this disease' which disease?

Author response: We are mentioning about PXF as disease in the sentence. The sentence is changed as per your suggestion in the revised manuscript.

Line 59 - which entity?

Author response: We are addressing PXF here as “this entity”. We have changed the sentence accordingly as per your suggestion in the revised manuscript.

Line 62 - what does 'This' refers to?

Author response: 'This' refers to PXF. We have added PXF in the sentence and rephrased it accordingly as per your suggestion.

Line 62-67 - Complex sentence. Please write simple sentences.

Author response: We have completely rephrased the sentences in the revised manuscript.

Line 69 - 'would be probably hold differences' - not clear

Author response: We want to say that there would be different proteins or markers expressed in different tissues of different stages of PXF disease that would be interesting to know.

Line 78 - what or which?

Author response: Typo error. We have changed in the revised manuscript.

Lines 103-104 - Write in simple sentences - The aim of the current study was...

Author response: We have changed the sentence accordingly as per your suggestion in the revised manuscript.

Line 105 - 'In brief' - this is rather explanatory than brief

Author response: We have changed the sentence accordingly as per your suggestion in the revised manuscript.

Line 120 - please specify if the diagnosis was done in this study

Author response: We have included “the diagnosis was done in this study” and changed the sentence accordingly as per your suggestion in the revised manuscript.

Line 153 - procedure for what was followed?

Author response: Procedure of the isolation of protein from the schirmer strip was followed. We have included about the procedure in the revised manuscript.

Line 170 - which guideline? Writing name of the manufacture in brackets will be good.

Author response: We have mentioned about the guidelines described in R&D System. R&D system is the manufacturer of the ELISA kit. 

Line 201 - 'examined examination' - correct the sentence

Author response: We have changed the sentence accordingly as per your suggestion in the revised manuscript.

Table 1 - Any reasoning for high male vs female numbers?

Author response: Since most of our patients of pseudoexfoliations are farmers by occupation, it is logical to expect more of male than females.

Line 312 - 'surprisingly' - state what was the expectation otherwise?

Author response: We wanted to say that the previous result of serum is consistent with the tears and aqueous humor of TGFβ1 ELISA results. There is a typo error we have changed the “surprising” word with “as expected” in the revise manuscript. 

Line 402 - 'Earlier studies' - mention the reference

Author response: I have added reference in the revised manuscript.

Line 436 - 'release of iris pigment release' - correct the sentence

Author response: We have changed the sentence accordingly as per your suggestion in the revised manuscript.

---

## [Editor Report · Decision Letter 1]

25 Mar 2021

TGFβ1, MMPs and cytokines profiles in ocular surface: Possible tear biomarkers for pseudoexfoliation

PONE-D-21-01128R1

Dear Dr. Rao,

We’re pleased to inform you that your manuscript has been judged scientifically suitable for publication and will be formally accepted for publication once it meets all outstanding technical requirements.

Kind regards,

Andrew W Taylor, Ph.D.

Academic Editor

PLOS ONE
---

## [Editor Report · Acceptance letter]

31 Mar 2021

PONE-D-21-01128R1 

TGFβ1, MMPs and cytokines profiles in ocular surface: Possible tear biomarkers for pseudoexfoliation 

Dear Dr. Rao:

I'm pleased to inform you that your manuscript has been deemed suitable for publication in PLOS ONE. Congratulations! Your manuscript is now with our production department. 

Kind regards, 

on behalf of

Dr. Andrew W Taylor 

Academic Editor

PLOS ONE